# IMPROVING REALISTIC SEMI-SUPERVISED LEARNING WITH DOUBLY ROBUST ESTIMATION

## ABSTRACT

A major challenge in Semi-Supervised Learning (SSL) is the mismatch between the labeled and unlabeled class distributions. Most successful SSL approaches are based on pseudo-labeling of the unlabeled data, and therefore are susceptible to confirmation bias because the classifier being trained is biased towards the labeled class distribution and thus performs poorly on unlabeled data. While distribution alignment alleviates this bias, we find that the distribution estimation at the end of training can still be improved with the doubly robust estimator, a theoretically sound approach that derives from semi-parametric efficiency theory. As a result, we propose a 2-stage approach where we first train an SSL classifier but only use this initial prediction for the doubly robust estimator of the class distribution, and then train a second SSL classifier but fixing the improved distribution estimation from the start. For training the classifier, we use a principled expectation-maximization framework for SSL with label shift, showing that the popular distribution alignment heuristic improves the data log-likelihood in the E-step, and that this EM is equivalent to the recent SimPro algorithm after reparameterization and logit adjustment but is much older and more interpretable (using the missingness mechanism). Experimental results demonstrate the improved class distribution estimation of the doubly robust estimator and subsequent improved classification accuracy with our 2-stage approach.

## 1 INTRODUCTION

Semi-supervised learning (SSL) aims to augment the small labeled set of data with a large unlabeled set of data Chapelle et al. (2009). This is of considerable practical significance since in many applications unlabeled data is easily available but the labeling effort is very costly. A key summary statistics is the *unlabeled* class distribution, which appeared in many previous semi-supervised algorithms (Kim et al., 2020; Wei et al., 2021; Oh et al., 2022; Wei & Gan, 2023; Du et al., 2024; Ma et al., 2024), and is identified as one of the main challenges in semi-supervised learning, namely when it is significantly different from the *labeled* class distribution. This is because most of these algorithms work by pseudo-labeling the unlabeled data Sohn et al. (2020); Berthelot et al. (2019a) and therefore are susceptible to confirmation bias (Arazo et al., 2020). The unlabeled class distribution can be of independent interest as well (Lee et al., 2025a;b), and can be used to adapt a classifier during *test time* (when unlabeled data is not available during training time as in semi-supervised learning), a procedure more commonly known as label shift adaptation (Saerens et al., 2002; Lipton et al., 2018; Azizzadenesheli et al., 2019; Alexandari et al., 2020)

In this paper, we first show that a simple yet principled expectation-maximization (EM) framework, which goes as far back as Ibrahim & Lipsitz (1996), underlies the popular distribution alignment heuristic in Berthelot et al. (2019a) and therefore is the foundation for most modern pseudo-labeling semi-supervised learning methods. SimPro (Du et al., 2024), a recent publication with very strong performance in various distribution mismatch settings, is in fact the same algorithm but with different parameterization using logit adjustment (see section 3.2). The framework shows that estimating the unlabeled class distribution is crucial for finding the pseudo-labels that improve the data log-likelihood in the E-step. However, we find that the final distribution estimation can still be improved, as evidenced in fig. 1, where SimPro tends to overestimate the head classes in 4 out of 5 unlabeled class distributions in their study, as compared to our proposed doubly-robust estimator. This leads to our second and main contribution, a simple 2-stage algorithm to improve learning further that

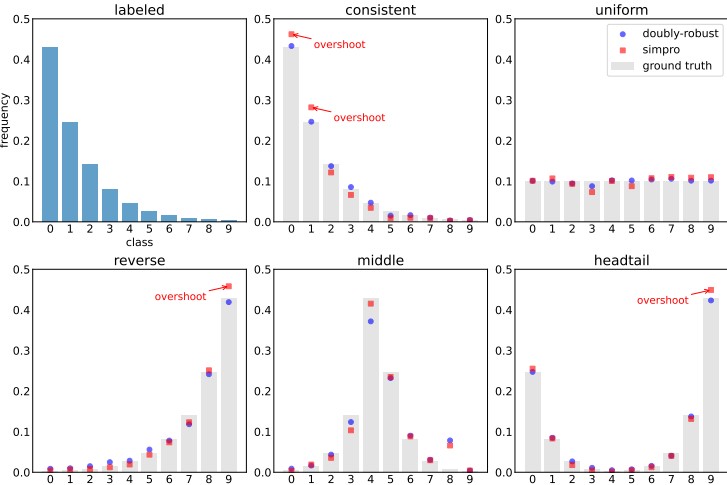

Figure 1: The labeled class distribution and 5 possible unlabeled class distributions studied in Du et al. (2024). SimPro significantly overestimates the head classes in consistent, reverse and headtail settings. Our doubly-robust estimate is more accurate at the head classes as well as the overall distribution in all but the middle setting, as measured in total variation distance in table 1. Our proposed 2-stage SimPro+ outperforms SimPro in classification accuracy in the middle setting as well.

first trains a model with EM, and then trains a second model also with EM but this time fixes the unlabeled data distribution with an improved estimate from the start.

Our doubly robust estimator derives from semi-parametric efficiency theory predominantly studied in causal inference and has well-understood and strong theoretical guarantee Chernozhukov et al. (2018). Its effectiveness stems from Neyman orthogonality, a statistical property that corrects first-stage estimation biases, including *but not limited* to SimPro's over-estimation (the theoretical conditions and results are stated in statistical convergence rates, see theorem 3.2). Our 2-stage approach first learns *nuisance* estimates including a first-stage classifier to use for the doubly robust estimator, a standard procedure in doubly machine learning with nuisance parameters (Chernozhukov et al., 2018). Experiments show that we improve the classification accuracy when using an improved unlabeled class distribution estimate in a second learning stage.

## 2 BACKGROUND AND RELATED WORK

**Notation** We write the random variable $X \in \mathcal{X}$ for the feature(s) and $Y \in \{1, \ldots, C\}$ for the class among $C$ possible classes. We are given a labeled dataset $D_l = \{x_i, y_i\}_{i=1}^{N_l}$ and an unlabeled dataset $D_u = \{x_i\}_{i=N_l+1}^{N}$, where $x_i$ and $y_i$ are realizations of $X$ and $Y$. The training dataset is $D_t = D_l \cup D_u$. The auxiliary variable $A$ takes binary values and selects between the different class distributions $P(Y|A)$; let $A = 1$ if the datapoint is in the labeled set and $A = 0$ in the unlabeled set. Therefore $P(Y|A = 0)$ is the class distribution the unlabeled set. The *combined* class distribution $P(Y) = P(A = 0)P(Y|A = 0) + P(A = 1)P(Y|A = 1)$ is the class distribution of the combined dataset. For convenience, we also denote $P(Y|unf) = 1/C$ everywhere to be the uniform class distribution, noting that it is not another value of $A$. We assume that the class distribution of the test set is uniform throughout this paper.

**Long-tailed Semi-supervised learning** is the intersection between long-tailed learning Buda et al. (2018) and semi-supervised learning Chapelle et al. (2009), and attempts to deal with two key real world problems: class distribution in the wild is often long-tailed with many classes having few samples; and the unlabeled data dwarfs the labeled data because of the advent of the web and the significant cost of large-scale manual labeling efforts. Pseudo labeling Lee et al. (2013); Berthelot et al. (2019b); Xie et al. (2020); Laine & Aila (2016) has become one of the prominent approaches in semi-supervised learning, and has been extended to the long-tailed case Wei et al. (2021); Lee et al. (2021), although the unlabeled class distribution was assumed to be the same as the labeled class distribution Berthelot et al. (2019a). More recent work has tackled the unknown distribution

case Zhao et al. (2022); Duan et al. (2022); Hu et al. (2022); Duan et al. (2023); Wei & Gan (2023); Du et al. (2024); Ma et al. (2024); Gan et al. (2024).

**(Balanced) Pseudo-labeling.** Semi-supervised learning methods use a regularization loss on the unlabeled data in addition to the classification loss on the labeled data. A simple technique is to use the model's own predictions on the unlabeled data. Specifically, FixMatch Sohn et al. (2020) keeps the maximum predictions that also fall above a certain confidence threshold and convert them into one-hot labels (operator $\delta$), which is called a pseudo label. For example, given a confidence threshold of $0.8$, a binary prediction $[0.1, 0.9]$ will be mapped to $[0, 1]$ while $[0.4, 0.6]$ to $[0, 0]$ under the operator $\delta$. FixMatch then minimizes the cross entropy loss between a strongly augmented version and the pseudo label of a weakly augmented version of the same unlabeled image:

$$L_u = - \sum_{i=N_l+1}^{N} \sum_{c=1}^{C} \delta(P(Y|\alpha(x_i)))_c \log P(Y = c|G(x_i)) \tag{1}$$

where $c$ is the class, $G$ is the strong augmentation, and $\alpha$ is the weak augmentation. FixMatch is simple and performant. However, it suffers when labeled and unlabeled class distributions are different, which label shift approach tries to address.

**Label shift and logit-adjustment.** Label shift assumes that the probability of $X$ given $Y$ is unchanged:

$$P(X|Y, A) = P(X|Y) \tag{2}$$

i.e. feature $X$ is conditionally independent of the variable $A$. The posterior change in $P(Y|X, A)$ results from the difference between the class distributions i.e. $P(Y|A = 0) \neq P(Y|A = 1)$. If the class distributions are known, logit adjustment can be used to convert a classifier of one class distribution to another. When label shift occurs between two datasets, classifiers performing well on one dataset may not perform well on the other. For example, to adapt the labeled class distribution $P(Y|A = 1)$ to the test class distribution $P(Y|unf)$, we can use Bayes formula to get:

$$P(Y|X, unf) \propto P(Y|X, A = 1) \frac{P(Y|unf)}{P(Y|A = 1)} \tag{3}$$

which is the basis of the post-hoc logit adjustment formula for long-tailed learning.

Label shift is the natural assumption in imbalanced / long-tailed learning where the target distribution is the uniform test distribution. Logit adjustment Menon et al. (2020), implicitly using this assumption, relies on the formula eq. (3) to correct label shift in long-tailed data. When the test distribution is unknown, label shift adaptation methods exist that can estimate the unknown test distribution Saerens et al. (2002); Alexandari et al. (2020); Lipton et al. (2018); Azizzadenesheli et al. (2019) when given a good classifier of the source data. It is possible therefore to train on the labeled set and use a label shift adaptation method to estimate the unlabeled class distribution. This procedure is best suited to label shift test-time adaptation Sun et al. (2023); Nguyen et al. (2024) where the unlabeled data is not available during model training. In contrast, when additional unlabeled data is available, semi-supervised EM gives significantly better class distribution estimation.

**Non-ignorable missingness.** This is a variant of missing data problems where the missing indicator $A$ can depend on both feature $X$ and outcome $Y$ Rubin (1976). The dependence on $Y$ distinguishes this variant from the standard ignorable missingness (missing at random) assumption Tsiatis (2006). The label shift assumption eq. (2) further assumes that only $Y$ causes $A$, and this assumption is sufficient to *identify* the true data distribution, meaning that no two distributions can generate our missing data Miller & Futoma (2023); Sportisse et al. (2023).

**Doubly robust (DR) estimation** This approach has roots in semi-parametric efficiency theory Kennedy (2024); Chernozhukov et al. (2018). The most successful application of DR is the estimation of the average treatment effect in causal inference Tsiatis (2006); Pham et al. (2023), which is an example of ignorable missingness. Recently doubly machine learning Chernozhukov et al. (2018; 2022) takes double robustness further by showing that powerful machine learning methods such as neural networks can be used to deal with high-dimensional and complex data while at the same time making valid inference about the target statistics. The applications of DR in modeling more complex data than traditionally studied in statistics have recently gained significant interest Shi et al. (2019); Chernozhukov et al. (2022); Zhang et al. (2023). We contribute to this line of work, but furthermore shows that we can plug in this estimation to improve the final classification itself.

More background and related work can be found in appendix C

## 3 METHOD

### 3.1 LABEL SHIFT EXPECTATION MAXIMIZATION

When pseudo-labeling is applied naively, a classifier trained on the labeled set with class distribution $P(Y|A = 1)$ may not do well on the unlabeled set that has a different class distribution $P(Y|A = 0)$ thus resulting in incorrect pseudo labels for training and consequently confirmation bias Arazo et al. (2020). We can not straightforwardly adapt to the unlabeled class distribution because it is unknown. In the following, we detail a likelihood maximization framework that eventually is shown to generalize pseudo-labeling to the label shift case. Using the indicator $A$, we can write the observed (or missing) data log-likelihood as

$$L(\theta) = \sum_{i=1}^{N_l} \log P(X = x_i, Y = y_i, A = 1|\theta) + \sum_{i=N_l+1}^{N} \log P(X = x_i, A = 0|\theta), \quad (4)$$

where $\theta$ represents the parameter of the joint distribution $P(X, A, Y)$. This likelihood consists of the labeled term and an unlabeled term with a missing $Y$. Immediately, we can maximize $L(\theta)$ by writing the unlabeled term as a $Y$-marginalization of the joint as in Sportisse et al. (2023). As we will use EM to maximize $L(\theta)$, we apply Jensen inequality to the each term in the second sum using the posterior weight $\omega^t(x, y) = P(Y = y|X = x, A = 0, \theta^t)$ where $\theta^t$ is value of $\theta$ in previous EM iteration, to get the lower bound

$$Q(\theta|\theta^t) = \sum_{i=1}^{N_l} \log P(X = x_i, Y = y_i, A = 1|\theta) + \sum_{i=N_l+1}^{N} \sum_{c=1}^{C} \omega^t(x_i, c) \log P(X = x_i, Y = c, A = 0|\theta)$$

$$(5)$$

This is the E-step of EM and we have found the "pseudo-label" $\omega^t(x, c)$ for our unlabeled data, reducing the problem to a supervised learning one for the moment. Now we need to decide how to decompose the joint $P(X, Y, A|\theta)$ which decides what the parameter specification will be. It is natural that we use the invariance $P(X|Y, A) = P(X|Y)$ in eq. (2) to decompose $P(X|Y)P(Y|A)P(A)$, but this requires generative modeling for $P(X|Y)$. Instead, we use $P(A|Y)P(Y|X)P(X)$, which means we only need to learn a classifier $P(Y|X)$ and a finite-dimensional $P(A|Y)$, which are recipes for the posterior weight $\omega^t(x, y)$. With this, we get

$$Q(\theta|\theta^t) = \sum_{i=1}^{N} \sum_{c=1}^{C} \gamma_i(c) \log P(Y = c|X = x_i, \theta) + \sum_{c=1}^{C} \sum_{a=0}^{1} \zeta_c(a) \log P(A = a|Y = c, \theta) \quad (6)$$

where $\gamma_i(c) = \mathbf{1}(y_i = c)$ for $i \leq N_l$ and $P(Y = c|X = x_i, A = 0, \theta^t)$ for $i > N_l$. $\zeta_c(1) = \sum_{i=1}^{N_l} \mathbf{1}(y_i = c)$ and $\zeta_c(0) = \sum_{i=N_l+1}^{N} P(Y = c|X = x_i, A = 0, \theta^t)$. This means that maximizing $L(\theta|\theta^t)$ is equivalent to minimizing a sum of cross entropy losses as this is the M-step. To compute the posterior weight $\omega^t(x, c)$, we use Bayes' theorem:

$$\omega^t(x, c) \propto P(Y = c|X = x, \theta^t)P(A = 0|Y = c, \theta^t) \quad (7)$$

In summary, the 2 steps of the EM are:

**E-step:** Given $P(Y = y|X = x, \theta^t)$ and $P(A = 0|Y = y, \theta^t)$, set $\omega^t(x, c)$ according to eq. (7)

**M-step:** Given $\omega^t(x, c)$, find the new $P(Y|X, \theta)$ and $P(A|Y, \theta)$ by maximizing $Q(\theta|\theta^t)$.

### 3.2 CONNECTION TO FIXMATCH, REMIXMATCH AND SIMPRO

Pseudo labeling methods such as Fixmatch (Sohn et al., 2020) and ReMixmatch (Berthelot et al., 2019a) have a deep connection with Expectation-Maximization. Indeed, eq. (1) without data augmentation and confidence thresholding is just the unlabeled term in eq. (5). For Fixmatch, $P(Y|A = 1) = P(Y|A = 0) = P(Y|uniform)$ i.e. there is no label shift. When there is mismatch, the distribution alignment heuristic in Berthelot et al. (2019a) and subsequent works use similar Bayes formula from eq. (3) to try to match the pseudo-label-induced class distribution with (an estimate of) the distribution of unlabeled class. This is equivalent to eq. (7) after observing that $P(A = 0|Y) \propto P(Y|A = 0)/P(Y)$. SimPro is a recent work that proposes an EM formula that we found to be equivalent this this EM up to a parameterization of the model and logit adjustment. They used a similar E-step but also applied Fixmatch's confidence thresholding and augmentation. Their M-step parameterizes the distribution as 2 parameters $\frac{P(X|Y)}{P(X)}$ and $P(Y|A = 0)$. This is just

another decomposition of the unlabeled log-likelihood term in eq. (5) up to a constant $P(A = 0)$:

$$\frac{P(X|Y)}{P(X)}P(Y|A=0) \propto \frac{P(Y|X)}{P(Y)}P(Y)P(A=0|Y) \tag{8}$$

Instead of canceling out $P(Y)$, however, SimPro uses a logit adjustment loss Menon et al. (2020) for the first term in eq. (6):

$$-\sum_{i=1}^{N}\sum_{c=1}^{C}\gamma_i(c)\log P(Y=c|X=x_i, unf, \theta)P(Y=c) \tag{9}$$

As $P(Y = c)$ is unknown, they use its running estimate. The model is automatically logit adjusted to the uniform test distribution during training. In contrast, if the model is $P(Y|X)$ in eq. (6), we can apply post-hoc logit adjustment. As shown in Menon et al. (2020), the logit adjustment loss is often slightly better, and this is what we find experimentally as well. Other than this difference, we can recover the class distribution $P(Y|A)$ from the missingness mechanism $P(A|Y)$ and because $P(A)$ is known, so they are learned equivalently.

### 3.3 OUR 2-STAGE ALGORITHM

Figure 2 (Appendix) shows an overview of our algorithm. In the first stage, we learn nuisance parameters to estimate the class distribution. This is a common procedure in Double Machine Learning Chernozhukov et al. (2018). We use the log-likelihood eq. (4) on a validation set to pick the final distribution. Then, we learn the final classifier in a second stage using the first-stage class distribution to adjust the pseudo-labels. The quality our first-stage estimation of $P(Y|A = 0)$ has a direct impact on the pseudo label accuracy, as highlighted in Theorem 3.1 of Wei et al. (2024). Briefly, the error gap between the adjusted model and the Bayes-optimal model can be bounded by the sum of an error term induced by the model's performance on the training data and another error term induced by the quality of our unlabeled distribution estimation. Therefore, we should aim for the highest estimation quality we can get in the first stage. To this end, we present 3 possible estimators for the combined class distribution $P(Y)$, the outcome regression (OR) estimator, inverse probability weighted (IPW) estimator and the doubly robust (DR) estimator. The unlabeled class distribution $P(Y|A = 0)$ can be recovered by noting that $P(Y) = \sum_a P(A = a)P(Y|A = a)$ and that $P(A)$ and the labeled class distribution $P(Y|A = 1)$ is known. The OR estimator is simply the average of the model's predictions

$$\Psi_{or}(c) = \frac{1}{N}\sum_{i=1}^{N}P(Y=c|X=x_i, \theta) \tag{10}$$

where the summation takes both labeled and unlabeled data.

Another estimator is the inverse probability weighted (IPW) estimator. Suppose that we have the ground truth missingness mechanism $P(A|Y)$, then we have the following identity:

$$P(Y=c) = \mathbb{E}_O\left[\frac{\mathbf{1}(A=1)}{P(A=1|Y)}\mathbf{1}(Y=c)\right] \tag{11}$$

where $O$ is a random variable representing one observation from the combined dataset, which is complete ($O = (X, A = 1, Y)$) if the datapoint is from the labeled set and missing ($X, A = 0$) if unlabeled set. The indicator $\mathbf{1}(A = 1)$ means that we are not actually using ground truth labels from the unlabeled set, but up-weighting the existing labels from the labeled set by the missingness mechanism. Replacing expectation with sample average and $P(A = 1|Y)$ with an estimation $P(A = 1|Y, \theta)$, we get our IPW estimator of $P(Y)$, which depends on $\theta$

$$\Psi_{ipw}(\theta)(c) = \frac{1}{N}\sum_{i=1}^{N}\frac{\mathbf{1}(a_i=1)}{P(A=1|Y=y_i, \theta)}\mathbf{1}(y_i=c) \tag{12}$$

**Our doubly robust estimator** It is worth noting that each estimator above (OR or IPW) uses only one part of the distribution, either $P(Y|X)$ or $P(A|Y)$. The DR estimator takes advantage of both of these quantities. It is

$$\Psi_{dr}(\theta)(c) = \frac{1}{N}\sum_{i=1}^{N}\left[P(Y=c|X=x_i, \theta) + \frac{\mathbf{1}(a_i=1)}{P(A=1|Y=y_i, \theta)}(\mathbf{1}(y_i=c) - P(Y=c|X=x_i, \theta))\right] \tag{13}$$

$\Psi_{dr}(\theta)$ is called doubly-robust because, given either a correct $P(Y|X)$ or $P(A = 1|Y)$, we will get an unbiased estimate of $P(Y)$. We need to learn both of these quantities from finite data which

Table 1: Performance of unlabeled distribution estimation methods on CIFAR-10-LT measured by Total Variation Distance to the ground truth. green indicates the best number in each column. In most settings, our SimPro + DR performs significantly better than other baselines.

| Model | Estimator | consistent | | uniform | | reversed | | middle | | head-tail | |
|---|---|---|---|---|---|---|---|---|---|---|---|
| | | $\gamma_l=150$ $\gamma_u=150$ | $\gamma_l=100$ $\gamma_u=100$ | $\gamma_l=150$ $\gamma_u=1$ | $\gamma_l=100$ $\gamma_u=1$ | $\gamma_l=150$ $\gamma_u=1/150$ | $\gamma_l=100$ $\gamma_u=1/100$ | $\gamma_l=150$ $\gamma_u=150$ | $\gamma_l=100$ $\gamma_u=100$ | $\gamma_l=150$ $\gamma_u=150$ | $\gamma_l=100$ $\gamma_u=100$ |
| Supervised | MLLS | $0.269 \pm 0.252$ | $0.038 \pm 0.006$ | $0.251 \pm 0.046$ | $0.255 \pm 0.060$ | $0.429 \pm 0.028$ | $0.493 \pm 0.050$ | $0.333 \pm 0.042$ | $0.320 \pm 0.009$ | $0.457 \pm 0.034$ | $0.444 \pm 0.043$ |
| Supervised | RLLS | $0.043 \pm 0.001$ | $0.044 \pm 0.010$ | $0.348 \pm 0.034$ | $0.305 \pm 0.068$ | $0.769 \pm 0.016$ | $0.678 \pm 0.028$ | $0.430 \pm 0.008$ | $0.368 \pm 0.013$ | $0.539 \pm 0.018$ | $0.503 \pm 0.020$ |
| MLE | IPW | $0.027 \pm 0.001$ | $0.027 \pm 0.000$ | $0.319 \pm 0.072$ | $0.243 \pm 0.010$ | $0.674 \pm 0.020$ | $0.646 \pm 0.041$ | $0.438 \pm 0.020$ | $0.454 \pm 0.026$ | $0.547 \pm 0.049$ | $0.491 \pm 0.059$ |
| MLE | OR | $0.045 \pm 0.004$ | $0.042 \pm 0.000$ | $0.215 \pm 0.026$ | $0.203 \pm 0.032$ | $0.433 \pm 0.017$ | $0.395 \pm 0.033$ | $0.193 \pm 0.006$ | $0.209 \pm 0.037$ | $0.307 \pm 0.147$ | $0.249 \pm 0.130$ |
| MLE | DR | $0.090 \pm 0.002$ | $0.079 \pm 0.000$ | $0.407 \pm 0.027$ | $0.360 \pm 0.007$ | $0.425 \pm 0.007$ | $0.421 \pm 0.029$ | $0.256 \pm 0.001$ | $0.286 \pm 0.031$ | $0.435 \pm 0.136$ | $0.362 \pm 0.122$ |
| EM | IPW | $0.035 \pm 0.002$ | $0.040 \pm 0.001$ | $0.021 \pm 0.001$ | $0.029 \pm 0.015$ | $0.303 \pm 0.187$ | $0.091 \pm 0.010$ | $0.119 \pm 0.011$ | $0.105 \pm 0.022$ | $0.104 \pm 0.026$ | $0.104 \pm 0.051$ |
| EM | OR | $0.037 \pm 0.003$ | $0.042 \pm 0.002$ | $0.016 \pm 0.001$ | $0.024 \pm 0.012$ | $0.269 \pm 0.183$ | $0.090 \pm 0.008$ | $0.122 \pm 0.012$ | $0.103 \pm 0.022$ | $0.072 \pm 0.012$ | $0.073 \pm 0.024$ |
| EM | DR | $0.034 \pm 0.004$ | $0.037 \pm 0.001$ | $0.014 \pm 0.001$ | $0.027 \pm 0.020$ | $0.264 \pm 0.191$ | $0.092 \pm 0.005$ | $0.111 \pm 0.019$ | $0.097 \pm 0.026$ | $0.077 \pm 0.016$ | $0.073 \pm 0.028$ |
| SimPro | IPW | $0.070 \pm 0.011$ | $0.058 \pm 0.000$ | $0.046 \pm 0.001$ | $0.049 \pm 0.005$ | $0.254 \pm 0.074$ | $0.223 \pm 0.098$ | $0.097 \pm 0.025$ | $0.067 \pm 0.002$ | $0.105 \pm 0.066$ | $0.110 \pm 0.079$ |
| SimPro | OR | $0.071 \pm 0.012$ | $0.058 \pm 0.000$ | $0.045 \pm 0.001$ | $0.049 \pm 0.006$ | $0.040 \pm 0.003$ | $0.059 \pm 0.017$ | $0.074 \pm 0.006$ | $0.075 \pm 0.002$ | $0.033 \pm 0.003$ | $0.033 \pm 0.003$ |
| SimPro | DR | $0.017 \pm 0.004$ | $0.026 \pm 0.001$ | $0.019 \pm 0.002$ | $0.018 \pm 0.003$ | $0.039 \pm 0.003$ | $0.058 \pm 0.025$ | $0.091 \pm 0.007$ | $0.031 \pm 0.001$ | $0.015 \pm 0.003$ | $0.019 \pm 0.007$ |

Table 2: Top-1 accuracy (%) on CIFAR-10-LT ($N_l = 500$, $M_l = 4000$) with different class imbalance ratios $\gamma_l$ and $\gamma_u$ under five different unlabeled class distributions. green / red indicates when our algorithm (SimPro+ and BOAT+) improves / degrades the base method (SimPro and BOAT). In most settings, our two stage algorithm improves SimPro (9 / 10) and BOAT (8 / 10)

| | consistent | | uniform | | reversed | | middle | | head-tail | |
|---|---|---|---|---|---|---|---|---|---|---|
| | $\gamma_l=150$ $\gamma_u=150$ | $\gamma_l=100$ $\gamma_u=100$ | $\gamma_l=150$ $\gamma_u=1$ | $\gamma_l=100$ $\gamma_u=1$ | $\gamma_l=150$ $\gamma_u=1/150$ | $\gamma_l=100$ $\gamma_u=1/100$ | $\gamma_l=150$ $\gamma_u=150$ | $\gamma_l=100$ $\gamma_u=100$ | $\gamma_l=150$ $\gamma_u=150$ | $\gamma_l=100$ $\gamma_u=100$ |
| FixMatch | $62.9 \pm 0.36$ | $67.8 \pm 1.13$ | $67.6 \pm 2.56$ | $73.0 \pm 3.81$ | $59.9 \pm 0.82$ | $62.5 \pm 0.94$ | $64.3 \pm 0.63$ | $71.7 \pm 0.46$ | $58.3 \pm 1.46$ | $66.6 \pm 0.87$ |
| CReST+ | $67.5 \pm 0.45$ | $76.3 \pm 0.86$ | $74.9 \pm 0.90$ | $82.2 \pm 1.53$ | $62.0 \pm 1.18$ | $62.9 \pm 1.39$ | $58.5 \pm 0.68$ | $71.4 \pm 0.60$ | $59.3 \pm 0.72$ | $67.2 \pm 0.48$ |
| DASO | $70.1 \pm 1.81$ | $76.0 \pm 0.37$ | $83.1 \pm 0.47$ | $86.6 \pm 0.84$ | $64.0 \pm 0.11$ | $71.0 \pm 0.95$ | $69.0 \pm 0.31$ | $73.1 \pm 0.68$ | $70.5 \pm 0.59$ | $71.1 \pm 0.32$ |
| Supervised | $63.2 \pm 0.14$ | $66.0 \pm 0.27$ | $63.3 \pm 0.28$ | $65.8 \pm 0.19$ | $63.1 \pm 0.19$ | $65.9 \pm 0.51$ | $63.5 \pm 0.22$ | $65.8 \pm 0.03$ | $63.0 \pm 0.18$ | $66.4 \pm 0.07$ |
| EM | $69.1 \pm 1.29$ | $73.8 \pm 0.71$ | $94.0 \pm 0.08$ | $93.2 \pm 0.94$ | $76.6 \pm 2.72$ | $82.2 \pm 0.24$ | $79.5 \pm 0.35$ | $81.6 \pm 0.58$ | $79.2 \pm 0.50$ | $79.8 \pm 0.17$ |
| SimPro | $74.4 \pm 0.71$ | $79.7 \pm 0.45$ | $93.3 \pm 0.10$ | $93.3 \pm 0.47$ | $83.8 \pm 0.80$ | $84.1 \pm 0.24$ | $78.7 \pm 0.30$ | $84.2 \pm 0.26$ | $81.2 \pm 0.20$ | $82.0 \pm 1.07$ |
| SimPro+ | $77.8 \pm 1.50$ | $81.2 \pm 0.39$ | $93.7 \pm 0.07$ | $93.7 \pm 0.24$ | $83.3 \pm 0.38$ | $84.7 \pm 0.78$ | $79.2 \pm 0.70$ | $85.4 \pm 0.66$ | $81.3 \pm 0.27$ | $82.5 \pm 0.56$ |
| BOAT | $80.5 \pm 0.39$ | $83.3 \pm 0.27$ | $93.9 \pm 0.03$ | $94.1 \pm 0.10$ | $79.7 \pm 0.25$ | $81.1 \pm 0.15$ | $79.7 \pm 1.15$ | $81.6 \pm 0.09$ | $79.4 \pm 0.44$ | $80.9 \pm 0.16$ |
| BOAT+ | $81.6 \pm 0.15$ | $83.8 \pm 0.04$ | $93.7 \pm 0.23$ | $94.1 \pm 0.17$ | $80.4 \pm 0.71$ | $81.7 \pm 0.38$ | $80.3 \pm 0.28$ | $83.1 \pm 0.45$ | $79.7 \pm 0.29$ | $81.0 \pm 0.36$ |

means their errors will propagate to the final estimation. However, this issue is addressed by the following optimality result.

### 3.3.1 THEORETICAL GUARANTEES FOR $\Psi_{dr}$

We can show, under weak assumption on the quality of $\theta$, that $\Psi_{dr}$ has strong theoretical guarantees. Let $o_p$ denote convergence in probability, define the $L_2(P)$ as $\|f\|_{L_2(P)} = (\int |f|^2 dP)^{1/2}$, where $P$ is the true distribution. We make the following assumption.

**Assumption 3.1.** Assume that both $P(Y|X,\theta)$ and $P(A=1|Y)$ converge at fourth-root-n rate i.e.
$$\|P(Y|X,\theta) - P(Y|X)\|_{L_2(P)} = o_p(N^{-1/4})$$
$$\|P(A=1|Y,\theta) - P(A=1|Y)\|_2 = o_p(N^{-1/4})$$
(14)

**Justification:** These assumptions (fourth-root-n rate of convergence) have been proven for neural networks Chernozhukov et al. (2022), which are consistent because of the universal approximation theorem, but tend to be biased because of regularization Chernozhukov et al. (2018).

We have the following optimality result:

**Theorem 3.2.** *Under the assumption theorem 3.1 the DR estimator $\Psi_{dr}$ for each class $c$ is asymptotically normal with 0-mean and the efficient influence function's variance:*
$$\sqrt{N}(\Psi_{dr}(\theta)(c) - P(Y=c)) \rightsquigarrow \mathcal{N}(0, \mathbb{E}[\phi(O)(c)^2])$$
(15)

The proof of theorem theorem 3.2 is deferred to the supplemental material. To put this theorem into perspective, the sample mean $\frac{1}{n}\sum_i z_i$ is the most efficient estimator of the mean of a random variable $Z$, where $z_i$ are unbiased samples. However, the OR estimator $\Psi_{or}$, which resembles a sample mean of $P(Y|X,\theta)$, can be biased because $\theta$ is a model-dependent approximation of the true data-generating process based on finite samples. This bias can slow the convergence of $\Psi_{or}$ if the model $P(Y|X,\theta)$ does not converge sufficiently quickly to the truth, specifically at a rate slower than $N^{-1/2}$ Chernozhukov et al. (2018), potentially causing $\Psi_{or}$ to diverge. Theorem 3.2 suggests that if both the OR and IPW estimators converge at a faster rate than $N^{-1/4}$, as stated in Assumption 3.1, then the DR estimator will converge to a normal distribution, behaving as if it

Table 3: Total Variance Distance on CIFAR-100-LT. SimPro + DR does not improve because there are too few labels for each class.

| | | consistent | | uniform | | reversed | | middle | | head-tail | |
|---|---|---|---|---|---|---|---|---|---|---|---|
| Model | Estimator | $\gamma_l = 20$ $\gamma_u = 20$ | $\gamma_l = 10$ $\gamma_u = 10$ | $\gamma_l = 20$ $\gamma_u = 1$ | $\gamma_l = 10$ $\gamma_u = 1$ | $\gamma_l = 20$ $\gamma_u = 1/20$ | $\gamma_l = 10$ $\gamma_u = 1/10$ | $\gamma_l = 20$ $\gamma_u = 20$ | $\gamma_l = 10$ $\gamma_u = 10$ | $\gamma_l = 20$ $\gamma_u = 20$ | $\gamma_l = 10$ $\gamma_u = 10$ |
| Supervised | MLLS | 0.707 ± 0.016 | 0.313 ± 0.100 | 0.445 ± 0.172 | 0.309 ± 0.119 | 0.383 ± 0.075 | 0.397 ± 0.006 | 0.570 ± 0.001 | 0.373 ± 0.107 | 0.543 ± 0.009 | 0.231 ± 0.057 |
| Supervised | RLLS | 0.520 ± 0.007 | 0.133 ± 0.003 | 0.337 ± 0.125 | 0.253 ± 0.082 | 0.424 ± 0.060 | 0.463 ± 0.003 | 0.454 ± 0.021 | 0.306 ± 0.074 | 0.460 ± 0.028 | 0.241 ± 0.040 |
| MLE | IPW | 0.075 ± 0.000 | 0.071 ± 0.001 | 0.229 ± 0.001 | 0.167 ± 0.002 | 0.565 ± 0.005 | 0.443 ± 0.007 | 0.415 ± 0.000 | 0.311 ± 0.005 | 0.343 ± 0.000 | 0.280 ± 0.001 |
| MLE | OR | 0.065 ± 0.002 | 0.061 ± 0.001 | 0.200 ± 0.007 | 0.143 ± 0.001 | 0.526 ± 0.011 | 0.399 ± 0.023 | 0.360 ± 0.003 | 0.256 ± 0.012 | 0.328 ± 0.003 | 0.266 ± 0.005 |
| MLE | DR | 0.149 ± 0.019 | 0.145 ± 0.010 | 0.243 ± 0.004 | 0.214 ± 0.019 | 0.568 ± 0.005 | 0.464 ± 0.014 | 0.403 ± 0.014 | 0.309 ± 0.012 | 0.365 ± 0.007 | 0.320 ± 0.004 |
| EM | IPW | 0.097 ± 0.008 | 0.092 ± 0.004 | 0.239 ± 0.007 | 0.179 ± 0.003 | 0.478 ± 0.012 | 0.329 ± 0.020 | 0.262 ± 0.016 | 0.202 ± 0.003 | 0.312 ± 0.002 | 0.227 ± 0.001 |
| EM | OR | 0.121 ± 0.007 | 0.108 ± 0.005 | 0.261 ± 0.007 | 0.189 ± 0.004 | 0.489 ± 0.013 | 0.335 ± 0.020 | 0.274 ± 0.016 | 0.211 ± 0.004 | 0.336 ± 0.003 | 0.235 ± 0.001 |
| EM | DR | 0.125 ± 0.005 | 0.111 ± 0.004 | 0.269 ± 0.007 | 0.194 ± 0.005 | 0.497 ± 0.010 | 0.336 ± 0.024 | 0.281 ± 0.019 | 0.219 ± 0.008 | 0.336 ± 0.007 | 0.233 ± 0.004 |
| SimPro | IPW | 0.125 ± 0.001 | 0.100 ± 0.005 | 0.166 ± 0.007 | 0.141 ± 0.009 | 0.353 ± 0.023 | 0.261 ± 0.008 | 0.202 ± 0.003 | 0.158 ± 0.005 | 0.277 ± 0.009 | 0.197 ± 0.003 |
| SimPro | OR | 0.133 ± 0.005 | 0.100 ± 0.004 | 0.160 ± 0.007 | 0.138 ± 0.010 | 0.322 ± 0.014 | 0.253 ± 0.008 | 0.202 ± 0.003 | 0.156 ± 0.005 | 0.269 ± 0.006 | 0.191 ± 0.004 |
| SimPro | DR | 0.122 ± 0.003 | 0.106 ± 0.006 | 0.188 ± 0.009 | 0.149 ± 0.006 | 0.343 ± 0.023 | 0.257 ± 0.007 | 0.219 ± 0.010 | 0.172 ± 0.002 | 0.279 ± 0.007 | 0.198 ± 0.004 |

Table 4: Top-1 accuracy (%) on CIFAR-100-LT. Despite poor estimation in stage 1, our algorithm does not degrade overall performance.

| | consistent | | uniform | | reversed | | middle | | head-tail | |
|---|---|---|---|---|---|---|---|---|---|---|
| | $\gamma_l = 20$ $\gamma_u = 20$ | $\gamma_l = 10$ $\gamma_u = 10$ | $\gamma_l = 20$ $\gamma_u = 1$ | $\gamma_l = 10$ $\gamma_u = 1$ | $\gamma_l = 20$ $\gamma_u = 1/20$ | $\gamma_l = 10$ $\gamma_u = 1/10$ | $\gamma_l = 20$ $\gamma_u = 20$ | $\gamma_l = 10$ $\gamma_u = 10$ | $\gamma_l = 20$ $\gamma_u = 20$ | $\gamma_l = 10$ $\gamma_u = 10$ |
| Supervised | 32.4 ± 0.40 | 38.4 ± 0.18 | 32.7 ± 0.25 | 38.0 ± 0.22 | 32.5 ± 0.51 | 38.4 ± 0.43 | 32.3 ± 0.08 | 37.9 ± 0.43 | 32.1 ± 0.33 | 38.2 ± 0.38 |
| EM | 42.4 ± 0.43 | 49.6 ± 0.30 | 50.9 ± 0.27 | 58.0 ± 0.35 | 42.1 ± 0.16 | 49.8 ± 0.47 | 42.8 ± 0.41 | 49.6 ± 0.36 | 41.5 ± 1.26 | 49.5 ± 0.18 |
| SimPro | 42.5 ± 0.58 | 49.6 ± 0.22 | 51.7 ± 0.22 | 58.1 ± 0.53 | 44.9 ± 0.21 | 51.8 ± 0.42 | 42.7 ± 0.06 | 49.8 ± 0.45 | 43.3 ± 0.76 | 50.9 ± 0.19 |
| SimPro+ | 42.8 ± 0.49 | 50.1 ± 0.33 | 51.6 ± 0.63 | 57.8 ± 0.38 | 44.7 ± 0.51 | 51.4 ± 0.88 | 43.4 ± 0.58 | 50.4 ± 0.28 | 43.8 ± 0.50 | 50.7 ± 0.76 |
| BOAT | 43.7 ± 0.16 | 51.4 ± 0.32 | 55.1 ± 0.95 | 60.5 ± 0.15 | 43.1 ± 0.49 | 52.7 ± 0.23 | 43.6 ± 0.19 | 51.4 ± 0.39 | 43.9 ± 0.42 | 51.4 ± 0.14 |
| BOAT+ | 44.8 ± 0.13 | 51.4 ± 0.51 | 53.8 ± 0.32 | 60.5 ± 0.69 | 43.4 ± 0.56 | 52.4 ± 0.36 | 43.9 ± 0.59 | 50.8 ± 0.09 | 43.6 ± 0.50 | 51.9 ± 0.49 |

were estimated using unbiased samples. Moreover, the DR estimator achieves the smallest variance among all regular estimators.

**Limitation of Theorem 3.2** This is an asymptotic result, guaranteeing convergence only as the sample size $N$ approaches infinity. In practice, for datasets with very small sample sizes per class, the DR estimator may offer little to no improvement over the baseline. Another weakness of the theorem is that it applies to each class $c$ independently; extending the result to all $C$ classes simultaneously would require roughly $C$ times as much data. Our experiments on CIFAR-100-LT confirm this behavior, showing that the DR estimator slightly degrades the baseline. However, the performance of the second-stage model remains relatively unaffected.

# 4 EXPERIMENTAL RESULTS

We show results for each stage of our algorithm. In the first stage, we compare among various methods to estimate the unlabeled class distribution $P(Y|A = 0)$, showing that SimPro + DR performs well. In the second stage, we freeze the unlabeled class distribution, using our best estimator SimPro + DR, and plug it into a second SimPro model (reseting model parameters). We also use alternative methods BOAT (Gan et al., 2024), the state-of-the-art at the time of writing, and ReMix-Match (Berthelot et al., 2019a), CoMatch (Li et al., 2021), the latter two are deferred to table 8 in the Appendix. We show that this simple procedure improves the existing SSL methods.

**Datasets** We evaluate our method on four standard semi-supervised learning benchmarks: CIFAR-10, CIFAR-100 Krizhevsky & Hinton (2009), STL-10 Coates et al. (2011), and Imagenet-127 Fan et al. (2022). To simulate RTSSL, we construct long-tailed labeled and unlabeled sets for CIFAR-10 and CIFAR-100. The labeled data follows an imbalance ratio $\gamma_l$ with head class size $n_1$, while the remaining class sizes are computed as $n_c = n_1 \times \gamma_l^{-\frac{c-1}{C-1}}$. The unlabeled data follows a similar setup with $\gamma_u$ and $m_1$. For CIFAR-10, we set $n_1 = 500$, $m_1 = 4000$, and test two configurations: $\gamma_l = \gamma_u = 150$ and $\gamma_l = \gamma_u = 100$. We generate 10 datasets by permuting the unlabeled class distributions in five ways: *consistent, uniform, reversed, middle*, and *head-tail*, as in Du et al. (2024). CIFAR-100 follows the same setup with $n_1 = 50$, $m_1 = 400$, and $\gamma_l, \gamma_u$ values of 20 and 10. For STL-10, where unlabeled data lacks ground-truth labels, we use all head-class samples and set $\gamma_l$ to 10 or 20. Imagenet-127 is naturally long-tailed with 127 classes, and we train on 32×32 and 64×64 resolutions as in Fan et al. (2022).

**Training.** We follow the implementation and hyperparameter settings of Du et al. (2024). We defer these details in the supplementary material. One important exception is that for Imagenet-127, we use the smaller Wide ResNet-28-2 in stage 1 and the larger ResNet-50 for stage 2, to demonstrate that a smaller model is sufficient for distribution estimation.

Table 5: Results on Imagenet-127. Each subtable reports a different metric.

(a) Total Variation Distance.

| Method | Estimator | $32 \times 32$ | $64 \times 64$ |
|---|---|---|---|
| MLE | IPW | $0.103 \pm 0.034$ | $0.051 \pm 0.000$ |
| MLE | OR | $0.153 \pm 0.052$ | $0.041 \pm 0.000$ |
| MLE | DR | $0.100 \pm 0.029$ | $0.075 \pm 0.003$ |
| EM | IPW | $0.141 \pm 0.006$ | $0.163 \pm 0.010$ |
| EM | OR | $0.205 \pm 0.006$ | $0.236 \pm 0.011$ |
| EM | DR | $0.024 \pm 0.001$ | $0.042 \pm 0.004$ |
| SimPro | IPW | $0.041 \pm 0.012$ | $0.224 \pm 0.040$ |
| SimPro | OR | $0.036 \pm 0.014$ | $0.291 \pm 0.079$ |
| SimPro | DR | $0.017 \pm 0.000$ | $0.037 \pm 0.004$ |

(b) Top-1 Accuracy (%).

| Method | $32 \times 32$ | $64 \times 64$ |
|---|---|---|
| SimPro | 54.8 | 63.7 |
| SimPro+ | 55.1 | 64.2 |
| BOAT | 51.6 | 58.7 |
| BOAT+ | 52.0 | 59.2 |

## 4.1 BETTER RESULTS ON LABEL DISTRIBUTION

We compare multiple methods for estimating the label distribution. Each method consists of a model, which is how the learning is done, and an estimator, which is how the final distribution is estimated using parameters learned from the model. The methods are: **Supervised** (using only supervised labels as in test-time adaptation) **MLE** (directly maximizing the likelihood eq. (4) with gradient descent Sportisse et al. (2023)); **EM** and **SimPro** as discussed in section 3.1 where the difference is that **EM** does not use confidence thresholding. While it is a powerful regularization technique that encodes the assumptions that classes are well separated Grandvalet & Bengio (2004), confidence thresholding can introduce bias to the label distribution estimation, as shown in fig. 1, which justifies the use of our doubly-robust estimator. Regarding estimators, for **Supervised**, we use **RLLS** Azizzadenesheli et al. (2019) and **MLLS** Saerens et al. (2002), both well-known in label shift adaptation, and for each semi-supervised method, we try all 3 estimators in section 3.3, namely **OR** which is the baseline estimator, **IPW** and **DR** which are our proposed estimator.

Results on table 1 presents the performance of various models and estimators on CIFAR-10. We can see that SimPro + DR performs best. In contrast, SimPro + OR, SimPro's original way of estimating $P(Y|A = 0)$, and SimPro + IPW tend to underperform EM on the consistent and uniform datasets. The consistent setting is worth noting, since it arises when data is sampled uniformly at random for labeling, representative of a large number of real world situations. EM is competitive to SimPro as well even without pseudo labeling, but overall we found this regularization to offer significant gains in the reversed, middle and head-tail settings. Finally, Supervised with either MLLS or RLLS estimators performs much worse than the semi-supervise methods.

table 3 shows that most methods struggle to estimate class distributions in CIFAR-100, which is expected given the limited label: only 50 labeled samples per class in the head class, compared to 500 in CIFAR-10, while the number of classes increases tenfold. Among SimPro variants, SimPro + OR performs best in most settings, though the performance gaps remain small. In contrast, on Imagenet-127, which has a similar number of classes but roughly ten times more samples, SimPro + DR shows significant improvements as shown in table 5a. As noted in theorem 3.2, our theoretical guarantee holds asymptotically and for each class independently, which may explain the limited effectiveness of our method in low-sample regimes. We ablate on the number of samples needed for DR to outperform OR by reducing the original CIFAR-10-LT size incrementally (there is not enough data to add for CIFAR-100-LT while fixing the class ratio), finding in fig. 3 (Appendix) that there is indeed a "phase transition" in the number of samples when DR starts to become better than OR. We also found that a small neural network and a small image resolution is sufficient for the distribution estimation of the much larger Imagenet-127 dataset.

## 4.2 TWO-STAGE ALGORITHM IMPROVES ACCURACY

In the second stage of our algorithm, we freeze our estimation and plug it in SimPro and BOAT. We denote SimPro+ and BOAT+ for algorithms that use our first stage estimate.

table 2 shows that for CIFAR-10 SimPro+ and BOAT+ improve over their original versions across most settings, leading to large improvements in both the consistent and middle class distribution settings. In particular, our two-stage approach improves SimPro in 9 / 10 settings and BOAT in 8 / 10 settings. We also observe consistent improvements over both base algorithms, SimPro and

Table 6: Top-1 Accuracy (%) on CIFAR-10. We compare our 2-stage SimPro+ with 1) an 1-stage alternative that updates and uses the doubly-robust estimation online and 2) SimPro+ with doubly-robust risk. We fix $\gamma_l = 150$. The chosen implementation outperforms alternatives.

| Method | consistent | uniform | reversed | middle | headtail |
|---|---|---|---|---|---|
| SimPro+ | 77.8 | 93.7 | 83.3 | 79.2 | 81.3 |
| batch-update | 71.9 | 91.4 | 82.6 | 78.6 | 81.2 |
| DR-risk | 72.1 | 89.8 | 67.1 | 75.6 | 79.5 |

BOAT, for several other datasets. table 7 (Appendix) demonstrates improvements for 2 / 2 class imbalance ratios in STL-10 and table 5b for 2 / 2 different resolutions of ImageNet-127.

We also evaluate on CIFAR-100 for multiple unlabeled class distribution settings and with mediocre class label distribution estimates in stage 1, demonstrate no degradation in accuracy in stage 2. As shown in table 4, the two stage algorithm with a mediocre stage 1 estimation leads to parity with the baseline. Stage 2 provides small improvements in 5 / 10 settings for SimPro and in 4 / 10 (with 2 ties) for BOAT.

### 4.3 ABLATION STUDY: ALTERNATIVE IMPLEMENTATIONS.

In this section, we ablate on our 2-stage choice. Specifically, we consider 2 alternative implementations:

**Doubly-robust risk**  This approach is Sportisse et al. (2023); Hu et al. (2022), as discussed in section 2. We consider the doubly-robust risk as our training loss eq. (28). The main difference is that instead of evaluating the class distribution $P(Y)$, the doubly robust risk evaluates the loss $l(X, Y|\theta)$ and therefore can be used directly as a training loss. Otherwise, the missingness mechanism and probability weight are shared by both. To control the comparison, we use the same missingness mechanism estimation from stage-1 for the doubly-robust risk, so the doubly-robust risk can be considered an alternative stage-2. More detail can be found in appendix C.

**Batch-update doubly-robust** $P(Y|A)$  One may ask why not combine the 2 stages into one by updating the class distribution with the doubly-robust estimator online. We verify this by re-computing the doubly-robust estimator for every batch. More specifically, after an M-step with a batch $t$ of data and we obtain $P(Y = y|X = x, \theta^t)$ and $P(A = 0|y = y, \theta^t)$, we plug these nuisance estimates into our doubly-robust estimator, still using the data from that batch only. To reduce variance, we then use a moving average of this newly computed value with a running value (a common strategy in SSL e.g. Berthelot et al. (2019a)), which is the actual value that we use for the posterior weight $\omega$ in the E-step.

table 6 shows that the both alternatives are worse in all mismatch settings, although the batch-update SimPro+ is better than the DR-risk. The batch-update doubly robust estimator does not have enough samples per batch for a reliable estimate, and the moving average has no theoretical backing and potentially introduces bias. Another alternative is update the doubly-robust estimator with all data but after every epoch, which even translates to multiple stages (instead of just 2) of updating the class distribution then updating the classifier. However, a crucial difference is that we reset the parameters after a stage, which alleviates a potential overfitting problem. The doubly-robust risk is worst overall, especially in the reversed setting where $P(A|Y)$ is very small for the labeled tail classes, causing instability issues during training. Another potential issue with the DR-risk is that the loss theoretically has undesirable optimum of negative infinity due to negative meta-pseudo-labels, see eq. (29). In conclusion, we find that the 2-stage implementation works best.

## 5 CONCLUSION

We improve the estimation of class distribution in the distribution mismatch settings in semi-supervised learning with the doubly robust estimator from semi-parametric efficiency theory, and subsequently propose a simple two-stage approach to additionally improve classification accuracy by training a second classifier using the improved class distribution. Our method achieves improved performance on three benchmark datasets: CIFAR-10, STL-10, and ImageNet-127. However, a limitation of our estimator is its reduced effectiveness on datasets with very small per-class sample sizes, such as CIFAR-100. A promising fix is to initialize the learning with large foundation models pretrained on image data, which is out of the scope of this work.

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

# A APPENDIX

# B PROOF OF THEOREM 3.2

We require some additional regularity assumptions:

**Assumption B.1.** 1) The number of classes $C$ is bounded w.r.t the number of samples $N$, 2) the missingness mechanism $P(A = 1|Y, \theta)$, as well as its estimated counterpart $P(A = 1|Y, \theta)$, are bounded below by some constant $\epsilon > 0$, 3) the quantities $P(Y|X, \theta)$ and $P(A|Y, \theta)$ are estimated using auxiliary samples independent of samples used for the sample averaging.

Assumptions 1 and 2 are natural. For the missingness mechanism, the ground truth being bounded means that there is a non-vanishing proportion of samples for every class. The boundedness of the estimate can be enforced by clipping the estimate. Assumption 3 is called sample splitting in Kennedy (2024).

For convenience we use operator $\mathbb{E}_N$ to denote the average of $N$ samples i.e. $\frac{1}{N}\sum_{i=1}^{N}$. Note that this is by itself a random variable, in contrast to $\mathbb{E}$ which is a fixed number.

*Proof of theorem 3.2.* Because $C$ is bounded (assumption B.1), we can fix a class $c$ and prove the theorem. Let us define the influence function $\phi$, parameterized by $\theta$, as

$$\phi(O|\theta)(c) = P(Y = c|X, \theta) + \frac{\mathbf{1}(A = 1)}{P(A = 1|Y, \theta)}(\mathbf{1}(Y = c) - P(Y = c|X, \theta)) - P(Y = c) \quad (16)$$

As we have done in the main text, we use $\phi(O)$ to denote the same function but all estimated quantities are replaced with their truths. In other words, we use $\phi(O)$ for $\phi(O|\theta_0)$ where $\theta_0$ is the truth, given that our model contains $\theta_0$ e.g. when the model is consistent.

Recall that:

$$\Psi_{dr}(\theta)(c) = \frac{1}{N}\sum_{i=1}^{N}\left\{P(Y = c|X, \theta) + \frac{\mathbf{1}(A = 1)}{P(A = 1|Y, \theta)}(\mathbf{1}(Y = c) - P(Y = c|X, \theta))\right\} \quad (17)$$

$$= \mathbb{E}_N[\phi(O|\theta)(c)] + P(Y = c)$$

We will show that:

$$\Psi_{dr}(\theta)(c) - P(Y = c) = (\mathbb{E}_N - \mathbb{E})[\phi(O)(c)] + o_P(N^{-1/2}) \quad (18)$$

To do that, we use the following decomposition

$$\Psi_{dr}(\theta)(c) - P(Y = c) = \mathbb{E}_N[\phi(O|\theta)(c)]$$

$$= (\mathbb{E}_N - \mathbb{E})[\phi(O)(c)] + (\mathbb{E}_N - \mathbb{E})[\phi(O|\theta)(c) - \phi(O)(c)] + \mathbb{E}[\phi(O|\theta)(c)] \quad (19)$$

and analyze the second and third term. The third term is:

$$\mathbb{E}[\phi(O|\theta)(c)] = \mathbb{E}[P(Y = c|X, \theta)] + \mathbb{E}\left[\frac{\mathbf{1}(A = 1)}{P(A = 1|Y, \theta)}(\mathbf{1}(Y = c) - P(Y = c|X, \theta))\right] - P(Y = c)$$

$$= \mathbb{E}\left[P(Y = c|X, \theta) + \frac{P(A = 1|Y)}{P(A = 1|Y, \theta)}(P(Y = c|X) - P(Y = c|X, \theta))\right] - \mathbb{E}[P(Y = c|X)]$$

$$= \mathbb{E}\left[(P(Y = c|X, \theta) - P(Y = c|X))(P(A = 1|Y, \theta) - P(A = 1|Y))\frac{1}{P(A = 1|Y, \theta)}\right] \quad (20)$$

by Cauchy-Schwarz inequality:

$$\mathbb{E}[\phi(O|\theta)(c)] \leq \frac{1}{\epsilon}\|P(A=1|Y,\theta) - P(A=1|Y)\|_2 \|P(Y=c|X,\theta) - P(Y=c|X)\|_{L_2(P)}$$

$$= \frac{1}{\epsilon}o_P(N^{-1/4}N^{-1/4}) = o_P(N^{-1/2})$$

(21)

by assumption 3.1 and that $P(A=1|Y,\theta) > \epsilon$ (assumption B.1). The second term can be bounded by Chebyshev inequality

$$P(|(\mathbb{E}_N - \mathbb{E})[\phi(O|\theta)(c) - \phi(O)(c)]| \geq t) \leq \frac{\mathbf{var}[\mathbb{E}_N[\phi(O|\theta)(c) - \phi(O)(c)]]}{t^2} = \frac{\mathbf{var}[\phi(O|\theta)(c) - \phi(O)(c)]}{Nt^2}$$

(22)

note here that $\theta$ is independent of the samples used for $\mathbb{E}_N$ by assumption B.1. For any $\varepsilon > 0$, by picking $t = \frac{1}{\sqrt{N\varepsilon}}$ we get

$$P\left(\left|\frac{(\mathbb{E}_N - \mathbb{E})[\phi(O|\theta)(c) - \phi(O)(c)]}{N^{-1/2}}\right| \geq \frac{1}{\sqrt{\varepsilon}}\right) \leq \varepsilon\mathbf{var}[\phi(O|\theta)(c) - \phi(O)(c)]$$

(23)

by the definition of $O_P$, we then get

$$(\mathbb{E}_N - \mathbb{E})[\phi(O|\theta)(c) - \phi(O)(c)] = O_P(N^{-1/2}\mathbf{var}[\phi(O|\theta)(c) - \phi(O)(c)])$$

(24)

Because $\phi$ is a continuous function of $P(Y|X,\theta)$ and $P(A|Y,\theta)$ (given $P(A|Y,\theta) > \epsilon$, assumption B.1), by the continuous mapping theorem and the fact that $P(Y|X,\theta)$ and $P(A|Y,\theta)$ are convergent in probability (assumption 3.1), we get $\mathbf{var}[\phi(O|\theta)(c) - \phi(O)(c)] = o_P(1)$. This gives

$$(\mathbb{E}_N - \mathbb{E})[\phi(O|\theta)(c) - \phi(O)(c)] = o_P(N^{-1/2})$$

(25)

Therefore, we have shown that the second and third term are both $o_P(N^{-1/2})$, proving eq. (18). As the final step, multiply both sides of this equation by $\sqrt{N}$ we get:

$$\sqrt{N}(\Psi_{dr}(\theta)(c) - P(Y=c)) = \sqrt{N}(\mathbb{E}_N - \mathbb{E})[\phi(O)(c)] + o_P(1) \rightsquigarrow \mathcal{N}(0, \mathbf{var}[\phi(O)(c)])$$

(26)

by the central limit theorem, and $\mathbf{var}[\phi(O)(c)] = \mathbb{E}[\phi(O)(c)^2]$ because $\mathbb{E}[\phi(O)(c)] = 0$. $\square$

While we started with the definition of $\phi$, eq. (18) shows that $\phi$ is indeed an influence function. Now we show that $\phi$ is also the efficient influence function, by using the characterization of the model's tangent space Tsiatis (2006). Note that the joint probability factorizes as $P(X,A,Y) = P(X)P(Y|X)P(A|Y)$, therefore the tangent space $\mathcal{T}$ factorizes as $\mathcal{T} = \mathcal{T}_X \oplus \mathcal{T}_{Y|X} \oplus \mathcal{T}_{A|Y}$ where $\mathcal{T}_X = \{h(X) : \mathbb{E}[h] = 0\}$, $\mathcal{T}_{Y|X} = \{h(X,Y) : \mathbb{E}[h|X] = 0\}$, $\mathcal{T}_{A|Y} = \{h(A,Y) : \mathbb{E}[h|Y] = 0\}$, and the 3 subspaces are pairwise orthogonal. All influence functions are orthogonal to the tangent space, but the influence function that is also in the tangent space has the smallest variance and is called the efficient influence function. As $\phi$ is already an influence function, we need only show that $\phi$ is in $\mathcal{T}$. We write $\phi$ as

$$\phi(O)(c) = (P(Y=c|X) - P(Y=c)) + \left[\frac{\mathbf{1}(A=1)}{P(A=1|Y)} - 1\right](\mathbf{1}(Y=c) - P(Y=c|X)) + (\mathbf{1}(Y=c) - P(Y=c|X))$$

(27)

and note that the first, second and third term are in $\mathcal{T}_X$, $\mathcal{T}_{A|Y}$ and $\mathcal{T}_{Y|X}$ respectively. Therefore, $\phi$ is indeed in $\mathcal{T}$. The efficient influence function has the smallest variance of all influence function, and therefore our estimator being asymptotically linear in $\phi$ (eq. (18)) has the smallest mean squared error in a local asymptotic minimax sense Kennedy (2024); Van der Vaart (2000)

## C  FURTHER BACKGROUND AND RELATED WORK

Our work builds and improves on Du et al. (2024). Specifically, we show in section 3.2 that it is a reparameterization of the semi-supervised EM algorithm in section 3.1, and we use it as the training method for both stages of our algorithm. Our work is also close to Sportisse et al. (2023); Hu et al. (2022) who also note the connection to non-ignorable missingness and propose doubly robust estimation of the loss. This loss remains consistent even when the pseudo labels are arbitrarily bad, in a similar spirit to Schmutz et al. (2022); Zhu et al. (2024), as long as the missingness mechanism is correct. Thus they try to safeguard against wrong un-adjusted labels. We on the other hand try to improve the label's quality via EM and adjustment by the doubly robust estimation of the unlabeled class distribution. An important weaknesses of the doubly robust loss Sportisse et al. (2023) is that it involves inverse-weighting Cui et al. (2019) which is prone to unstable training Ren et al. (2020).

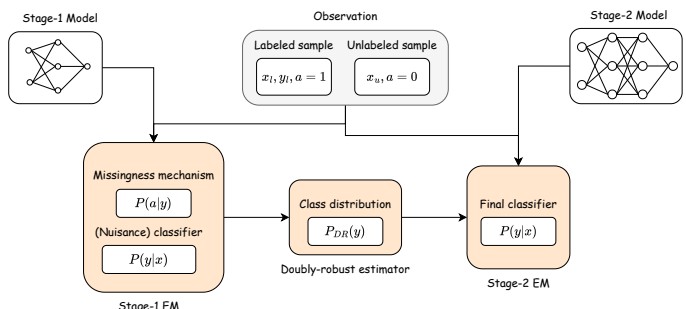

Figure 2: Overview of our 2-stage method (section 3.3). In stage 1, we use Expectation-Maximization (EM, section 3.1) to estimate the missingness mechanism and classifier from observable data. These quantities are used as nuisance components for the doubly-robust estimator of the class distribution eq. (13). In stage 2, we can use EM or other existing methods that also use logit-adjustment with the (unlabeled) class distribution to estimate the final classifier. We use SimPro as our implementation of EM (section 3.2). The network in stage 1 can be of equal or smaller size than the network in stage 2 (section 4.1).

**Discussion on semi-supervised EM.** It appears that semi-supervised EM was first used for parameter estimation when the missingness mechanism is non-ignorable in Ibrahim & Lipsitz (1996), but has not been used for label shift estimation. Perhaps this is because the semi-supervised situation where additional unlabeled data is available during training is rarer than the test-time adaptation case. EM is well suited to take advantage of the extra unlabeled data to improve the classifier under very scarce and long-tailed labeled data. While the connection between pseudo-labeling and EM has been explored before Grandvalet & Bengio (2004), the situation with label shift has not until recently Du et al. (2024). Here the application of EM is much more interesting, because other than simply giving pseudo-labeling a rigorous formulation, EM also estimates the missingness mechanism (equivalently the label distribution shift), which is important for shift correction and thus high-quality pseudo-labels Wei & Gan (2023). The application of confidence thresholding can be seen as a sparse variant of EM Neal & Hinton (1998).

**The doubly-robust risk.** A technique that also derives from the theory of semi-parametric efficiency is orthogonal statistical learning (Foster & Syrgkanis, 2023). The idea is to minimize the doubly-robust risk:

$$\mathcal{R}(\theta_2) = \frac{1}{N} \sum_{i=1}^{N} \left[ l(x_i, \hat{y}_i | \theta_2) + \frac{\mathbf{1}(a_i = 1)}{P(A = a_i | Y = y_i, \theta_1)} (l(x_i, y_i | \theta_2) - l(x_i, \hat{y}_i | \theta_2)) \right] \quad (28)$$

where $l(x, y | \theta) = -\sum_{c=1}^{C} [y]_c \log P(Y = c | X = x, \theta)$ is the negative cross-entropy. The notation $[y]_c$ means that we are using the $c$-entry in a C-dimension probability vector $y$. Thus, $y_i$ denotes the one-hot label of observation $i$, while $\hat{y}_i$ denotes the pseudo-label, which can be one-hot or all-zero. Finally, we use $\theta_1$ to denote that $P(a|y, \theta_1)$ is an estimation from a previous stage, but it can be estimated with $\theta_2$ as well. The risk $\mathcal{R}(\theta_2)$ can be used as a training loss in a straightforward fashion. Similar to the doubly robust estimation of $P(Y)$, the doubly robust risk provides approximately unbiased estimation of the risk. This property has been used in (Sportisse et al., 2023; Hu et al., 2022; Zhu et al., 2024) also in the semi-supervised learning setting. More broadly, it is at the heart of one of the core techniques in heterogenous treatment effect estimation in causal estimation Kennedy (2023); Foster & Syrgkanis (2023); Wager & Athey (2018). The focus here is not the estimation of $\mathcal{R}(\theta_2)$ per se, but the quality of the learned model Foster & Syrgkanis (2023). By using the doubly-robust risk, we can achieve an optimality result similar in spirit to our theorem theorem 3.2, but for the generalization error. While this is appealing, in practice there are 2 problems with this approach. First, the inverse probability weight $P(A = a_i | Y = y_i, \theta_1)$ can be very large if the class ratio is highly unlabeled, making training unstable Kallus (2020); Pham et al. (2023). This problem exists for our estimation as well. However, it is much easier to control for estimation than for training

Table 7: Top-1 Accuracy (%) on STL-10. Our two-stage algorithms improves both SimPro and BOAT for both settings.

| Method | $\gamma_l = 10$ | $\gamma_l = 20$ |
|---|---|---|
| Supervised | $73.9 \pm 0.57$ | $70.4 \pm 0.95$ |
| MLE | $67.6 \pm 0.57$ | $58.9 \pm 4.05$ |
| EM | $84.9 \pm 0.14$ | $83.6 \pm 0.25$ |
| SimPro | $82.4 \pm 1.57$ | $80.5 \pm 0.96$ |
| SimPro+ | $83.9 \pm 0.76$ | $82.7 \pm 0.86$ |
| BOAT | $83.8 \pm 0.20$ | $82.0 \pm 0.34$ |
| BOAT+ | $84.1 \pm 0.38$ | $82.4 \pm 0.10$ |

because of the iterative nature of model update. Secondly, we can further write $\mathcal{R}$ as:

$$\mathcal{R}(\theta_2) = \frac{1}{N} \sum_{i=1}^{N} l \left( x_i, \hat{y}_i + \frac{\mathbf{1}(a_i = 1)}{P(A = a_i | Y = y_i, \theta_1)} (y_i - \hat{y}_i) \Big| \theta_2 \right) \qquad (29)$$

which is a cross-entropy loss with new meta-pseudo-labels. However, these labels are not meant to be learned exactly, and furthermore they can be negative. Thus, theoretical works have to put stringent assumptions on the models. In section 4.3, we show that experimentally that the instability problem makes doubly-robust risk performance worse than our 2-stage approach.

## D  TRAINING AND HYPERPARAMETER SETTINGS.

For neural network training, we follow the implementation and hyperparameter settings of Du et al. (2024). In particular, we adapt the core code of SimPro for Supervised, MLE and EM. For MLE, we update $P(A|Y)$ using the Adam optimizer with learning rate 1e-3, while for EM we use a momentum update similar to SimPro's update of $P(Y|A)$ because it has a a closed-form solution at each mini-batch. We use Wide ResNet-28-2 on all methods and all datasets in this section, including Imagenet-127, because we are motivated by the fact that stage-1's goal is not classification accuracy but the estimation of a finite-dimensional parameter. When using Wide ResNet-28-2 for Imagenet-127, we use the hyperparameters of CIFAR-100, except we lower the batch size of unlabeled data to 2 times that of labeled data instead of 8 for memory reason. We do not perform additional hyperparameter tuning. All experiments can be performed on 1 A6000 RTX GPU, and are run 3 times. We report the total variation distance between the estimated and the ground truth unlabeled class distribution, similar to its usage in Theorem 3.1 of Wei et al. (2024), and the top-1 classification accuracy.

In the second stage of our algorithm, we freeze our estimation and plug it in SimPro and BOAT. We keep exactly the same hyperparameter settings that SimPro and BOAT use. In particular, for Imagenet-127, we now use ResNet-50 and run each experiment once. In SimPro, we set the un-labeled class distribution $P(Y|A = 0)$ at the E-step; however, we still keep a running estimate of the class distribution $P(Y)$ in the logit adjustment loss eq. (9). While it is possible to use the first stage estimate in the logit adjustment loss, we observe that doing so results in lower accuracy than using the the running average. This is conceptually consistent with the role of the running average - serving not as an accurate estimate of $P(Y)$ but to make the classifier's class distribution uniform through the logit adjustment loss, which is good for the test set. Similarly, in BOAT, we only replace $\Delta_c = \log P(Y|A = 1) - \log P(Y|A = 0)$ in equation (4) of Gan et al. (2024), which is adjusting a classifier's predictions from the labeled to the unlabeled class distribution, with our SimPro + DR estimate instead of their on-the-fly estimate.

## E  ADDITIONAL EXPERIMENTS

Table 8 (supplementing Table 3) presents results for two additional methods, ReMixMatch Berthelot et al. (2019a) and CoMatch Li et al. (2021). Methods augmented with DR show significant improvements, except in one setting where performance is comparable.

Figure 3 simulates scenarios with few samples relative to the number of classes. We vary the sample size, holding everything else constant including the imbalance ratio, and find that there is indeed a

Table 8: Top-1 accuracy (%) on CIFAR-10-LT ($N_l = 500$, $N_u = 4000$). This table supplements table 2 from the main paper by including results for ReMixMatch and CoMatch.

|  | consistent | uniform | reversed | middle | head-tail |
|---|---|---|---|---|---|
|  | $\gamma_l = 150$ $\gamma_u = 150$ | $\gamma_l = 150$ $\gamma_u = 1$ | $\gamma_l = 150$ $\gamma_u = 1/150$ | $\gamma_l = 150$ $\gamma_u = 150$ | $\gamma_l = 150$ $\gamma_u = 150$ |
| ReMixMatch Berthelot et al. (2019a) | 64.5 | 50.9 | 48.9 | 43.9 | 52.2 |
| ReMixMatch+DR | 63.6 | 80.7 | 50.4 | 48.2 | 70.28 |
| CoMatch Li et al. (2021) | 70.8 | 76.9 | 65.8 | 58.4 | 65.3 |
| CoMatch+DR | 73.1 | 76.7 | 75.3 | 68.8 | 75.0 |

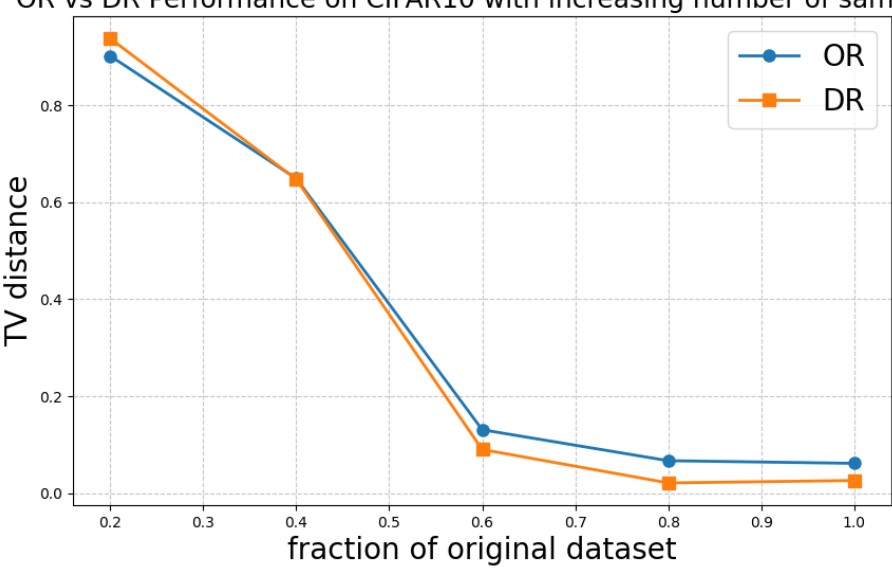

Figure 3: Effect of sample size fraction ($x$) on DR performance relative to OR (baseline). Dataset: CIFAR-10 with $N_l = 500x$, $N_u = 4000x$, $\gamma_l = 100$, $\gamma_u = 100$. DR performance is comparable to or worse than OR for $x \leq 0.4$.

"phase transition" in the number of samples at 0.4 fraction of the original dataset when DR starts to become better than OR at estimating the class distribution.

