# OpenReview forum: "Improving realistic semi-supervised learning with doubly robust estimation"
_ICLR.cc/2026/Conference — Submitted to ICLR 2026_

### Official Review · Reviewer_hNVB · 2025-10-26

**Soundness:** 3
**Presentation:** 3
**Contribution:** 2
**Rating:** 4
**Confidence:** 3

**Summary:**

This paper addresses realistic semi-supervised learning (RTSSL), where the labeled and unlabeled sets follow different long-tailed class distributions. It proposes a two-stage procedure that first estimates the unlabeled class distribution and then plugs this estimate into existing SSL methods. In Stage 1, the paper estimates P(Y) via a doubly robust (DR) estimator that combines outcome regression (OR; averaging predictions) and inverse probability weighting (IPW). Under Assumption 3.1 (e.g., N^{1/4} rates for the nuisance models P(Y|X) and P(A|Y)), it shows per-class asymptotic normality at the semiparametric efficiency bound (Theorem 3.2). In Stage 2, the estimated distribution is frozen and used to retrain SimPro/BOAT. On CIFAR-10, STL-10, and ImageNet-127, TVD improves and top-1 accuracy increases modestly. On CIFAR-100, where per-class sample sizes are small, DR is less advantageous in Stage 1, but the two-stage training generally preserves accuracy. Batch-wise online DR updates and single-stage “DR-risk” training are less stable or underperform.

**Strengths:**

* **Originality**:
  The paper reframes SimPro and distribution alignment within a missing-data + EM formulation with label-dependent selection, clarifying how confidence thresholding can bias distribution estimates and motivating DR in the estimation step.

* **Quality**:
  While the DR form is standard (OR + IPW), Assumption 3.1 and Theorem 3.2 are stated clearly, and the efficiency claim is carefully argued. The two-stage implementation is simple and reproducible; comparisons against online updates and ``DR-risk'' are included and support the chosen design.

* **Clarity**:
  The evaluation separates distribution estimation---reported as total variation distance (TVD)---from downstream classification (top-1). Stage 2 freezes the estimated distribution, and the figures/tables make the head-class overestimation and its correction clear.

* **Significance**:
  The recipe can be dropped into existing SSL pipelines (SimPro/BOAT) with little engineering. Improvements on ImageNet-127 indicate practical value, and Stage~1 can use a small backbone, which is operationally attractive.

**Weaknesses:**

* **Limited methodological novelty.**:
  The estimator follows the standard doubly robust template (outcome regression + inverse-probability weighting) justified by a conventional influence-function argument. Its form and assumptions match established semiparametric results; distinctiveness mainly comes from the RTSSL setting and freezing the estimated distribution in a two-stage pipeline. No new weighting, orthogonalization, or debiasing mechanics are introduced, so the methodological advance is incremental.

* **Scope of theory.**:
  The efficiency claim is per class and asymptotic, relying on N^(1/4) rates for nuisance estimators and positivity-type conditions. These guarantees are least informative with many classes and sparse counts; finite-sample bounds for the full distribution (e.g., TVD) and simultaneous control across classes are not provided. Sensitivity to misspecification in P(A|Y) and variance inflation from inverse weights is not quantified; weaker gains on CIFAR-100 are consistent with these limits.

* **Experimental scale.**:
  Experiments use CIFAR-10/100, STL-10, and low-res ImageNet-127—small/medium by modern SSL standards—with modest Stage 1 backbones and 224 px not evaluated. Unlabeled pools come from the same datasets, so web-scale shift/noise is lightly represented. Improvements are generally modest and may be dataset-dependent, limiting external validity without larger or higher-resolution studies.

**Questions:**

1. **Finite-sample error assessment.**:
As stated, Theorem 3.2 gives per-class, asymptotic guarantees, so behavior in multi-class, small-sample regimes remains unclear. It would be valuable to comment on finite-sample scaling with respect to the number of classes, the clipping level, and nuisance-estimation error. If deriving a non-asymptotic bound is beyond scope, please include an empirical calibration—for example: total variation distance (TVD) and the maximum per-class error plotted against estimated per-class effective sample size, simultaneous bootstrap intervals across classes, and a sensitivity analysis to misspecification in P(A|Y).

2. **Behavior under scale-up.**:
The current evidence appears limited to small–medium scale. To assess external validity, please increase the input resolution in Stage-1 while keeping Stage-2 fixed, and report head/middle/tail breakdowns, calibration (ECE/NLL), and runtime/compute. This would help clarify whether the DR estimator remains preferable as P(Y|X) improves.

3. **Switching criterion between OR and DR**:
The empirical trends suggest a data-dependent, phase-like transition between OR and DR. It would help to state a practical threshold—such as a per-class labeled or effective sample size (possibly incorporating estimated P(A|Y) and class prevalence)—beyond which DR tends to outperform OR in TVD. Please also provide a simple pre-training diagnostic (required inputs, a few probing steps, and a decision rule), and validate via accuracy/TVD changes near the threshold for both head and tail classes.

4. **Failure modes by mismatch type.**:
   Using existing results, please summarize per-pattern TVD together with concise indicators of classifier and selection-model errors, and add a brief sensitivity discussion (holding one nuisance fixed). This is expected to clarify the dominant source of error in each regime and the conditions under which DR is likely to have an advantage.

---

### Official Review · Reviewer_NNjf · 2025-10-29

**Soundness:** 2
**Presentation:** 2
**Contribution:** 2
**Rating:** 4
**Confidence:** 4

**Summary:**

This paper attempts to tackle the problem of class distribution mismatch between labeled and unlabeled data in SSL. The authors claim that existing pseudo-labeling methods, such as SimPro, suffer from confirmation bias, leading to poor estimates of the unlabeled class distribution. Their core idea is a two-stage approach: first, they train a standard SSL classifier; second, they use the outputs of this model to obtain improved estimate of the unlabeled class distribution via a doubly robust estimator; finally, they reset the model parameters and train a second SSL classifier from scratch using this fixed, improved distribution. Experimental results demonstrate that this two-stage method improves classification accuracy on several benchmarks.

**Strengths:**

1. The paper does address an important and practical problem in SSL. The assumption that unlabeled data follows the same distribution as the small labeled set is often violated in the real world.

**Weaknesses:**

1. The core contribution of this paper is negligible. The proposed method is essentially a brute-force and uninspired plug-and-play application of the DR estimator onto an existing SSL pipeline.
2.  After first stage, why train a full model, only to discard it entirely and train another one from scratch? Maybe authors can train from the model from the first stage.
3. The authors should provide a detailed comparison of the computational cost. The method requires training a model twice, in full. If the total training time is double that of the baseline, of what practical significance is the marginal accuracy gain on CIFAR-10?

**Questions:**

See in Weaknesses.

---

### Official Review · Reviewer_YPut · 2025-10-31

**Soundness:** 2
**Presentation:** 3
**Contribution:** 2
**Rating:** 4
**Confidence:** 5

**Summary:**

This paper addresses the bias caused by class distribution mismatch between labeled and unlabeled data in semi-supervised learning (SSL). The authors propose a two-stage training framework: in the first stage, a doubly robust estimator derived from semi-parametric theory is used to estimate the class distribution; in the second stage, a new classifier is trained using the improved distribution obtained from the first stage. The paper further provides an EM-based theoretical interpretation of the method and demonstrates its effectiveness on four public datasets.

**Strengths:**

- The related work section is comprehensive and provides solid background knowledge on class imbalance in SSL.
- The proposed method is supported by a clear and rigorous theoretical foundation.
- Experimental results demonstrate the superiority of the method from multiple perspectives, including distribution alignment accuracy and final classification performance.

**Weaknesses:**

- The paper’s motivation relies on a severe label–unlabeled distribution mismatch constructed through artificially long-tailed label splits. However, it remains unclear how representative such an extreme imbalance is in real-world semi-supervised learning (SSL) scenarios. Besides, class imbalance is often mitigated using balanced or reweighted strategies. Comparing against such class-balanced baselines—e.g., FixMatch with balanced distribution alignment on SVHN or CIFAR-LT—would provide stronger justification that the proposed DR approach offers unique advantages beyond standard balancing methods. Personally, SVHN, although not perfectly balanced, can achieve notable improvements when balanced distribution alignment is applied, without requiring the additional estimation.
- The motivation behind the proposed approach is not clearly articulated. The paper only mentions that SimPro tends to overestimate head classes and that the doubly robust estimator is more accurate, but it remains unclear what the fundamental underlying problem is. Is this a general limitation shared by existing SSL methods, or a specific weakness of SimPro-style training?
- DR estimators and label-shift EM are well-established; the contribution is mainly engineering the DR estimate into a two-stage SSL pipeline. The conceptual gap over prior SimPro/EM-style SSL is modest.
- Theoretical analysis seems to rely on strong conditions like bounded support, fourth-root-n convergence, and correct model specification. What if the assumptions cannot be satisfied?
- The experiments are conducted on several small-scale benchmark datasets, which limits the practical relevance of the results. Evaluation on larger and more diverse datasets would strengthen the claims.
- Authors are encouraged to inlcude more recent SSL methods and more experimental results like robustness analysis under varying pseudo-label confidence thresholds, DR estimator variance (IPW instability).
- A large portion of the paper is devoted to background and related work, while the core methodological description is relatively brief.

**Questions:**

The paper appears to rest on the premise that all we need is to estimate accurate distribution for imbalanced SSL, and SSL performance will reliably improve if this distribution is estimated correctly. Please carefully explain such premise, discuss the realism of these severe long-tailed SSL setting, clarify the core motivation.

---

### Official Review · Reviewer_jQfq · 2025-11-01

**Soundness:** 2
**Presentation:** 3
**Contribution:** 2
**Rating:** 4
**Confidence:** 3

**Summary:**

*   **Formulation:** Frames Semi-Supervised Learning (SSL) as a missing-data problem under label shift.
*   **Method:** A two-stage EM-based procedure that:
    1.  Estimates the class prior (π) using robust estimators (OR, IPW, DR).
    2.  Applies a logit-adjusted pseudo-labeling strategy for final model training.
*   **Theoretical Result:** The Doubly Robust (DR) estimator is proven to be asymptotically efficient, even under $O(N^{-1/4})$ convergence rates for nuisance parameters.
*   **Empirical Result:** The approach demonstrates improved class-prior estimation and yields consistent, modest accuracy gains on real-world SSL benchmarks.

**Strengths:**

The DR estimator for the finite-dimensional class prior is doubly robust and semiparametrically efficient under standard conditions, offering a clean theory-to-practice bridge absent in many RTSSL works.

**Weaknesses:**

## Weaknesses

### 1. Fragile Identification under Label-Shift Assumption
The paper’s identification strategy critically depends on the *label-shift* assumption—i.e., \( P(X|Y) \) remains invariant across labeled and unlabeled data, while \( P(Y) \) may change. However, this assumption is often violated in realistic semi-supervised or weakly supervised settings where labeling propensity depends on both class and features.

When selection depends on \( X \) given \( Y \)—for instance, when more prototypical or high-quality examples are labeled first while ambiguous or hard-to-classify samples remain unlabeled—the inverse-propensity weights \( 1 / P(A=1|Y) \) and corresponding doubly robust (DR) corrections become unstable or ill-defined for certain feature regions. This limits the method’s robustness to practical data collection biases.

> **Overall:** The proposed estimator’s performance and validity may deteriorate substantially under mild violations of label shift.

---

### 2. Incomplete Evaluation Metrics and Diagnostic Analysis
The empirical validation focuses solely on total variation (TV) distance between estimated and true class priors. While TV is a reasonable and interpretable measure for discrete priors, it fails to capture sensitivity to small but impactful deviations in tail classes or low-probability regions.

Measures like KL divergence (log-likelihood loss) would provide complementary insight, especially given that small errors in low-frequency classes can cause large effects downstream. The absence of KL, per-class errors, and calibration diagnostics weakens the evaluation’s comprehensiveness and makes it difficult to interpret why “Stage-2” improvements are modest.

> **Overall:** The evaluation provides limited evidence of estimator quality across different regimes (e.g., balanced vs. long-tailed).

**Questions:**

## Questions for the Authors

1. **Assumption robustness**
   - How sensitive is the DR estimator to mild violations of the label-shift assumption?
   - Have the authors explored reweighting or modeling strategies that allow partial dependence on \( X \) in \( P(A=1|Y, X) \)?
   - Could the instability caused by near-zero propensities be mitigated through truncation, regularization, or targeted estimation methods?

2. **Empirical diagnostics**
   - Why was TV distance chosen as the sole evaluation metric?
   - Would the conclusions change if KL divergence or per-class calibration metrics were reported?
   - Can the authors provide diagnostic plots showing calibration of the estimated priors (e.g., reliability diagrams or class-wise bias)?

3. **Stage-2 performance**
   - What factors explain the modest improvements in Stage-2?
   - Could adding KL-based optimization or regularization improve Stage-2 adaptation?

---

### Meta-Review · Area_Chair_ZqAg · 2025-12-29

**Summary:**

This work studies the research problem on class distribution mismatch between labeled and unlabeled data in semi-supervised learning. The motivation is the observation that pseudo-labeling approaches often overestimate head classes. This work further proposed a two-stage framework that first estimates the unlabeled class distribution using a doubly robust estimator, and then retrains an SSL model with this fixed, improved distribution. Experiments on several SSL benchmarks show modest but consistent improvements.

The main strengths are that (1) the research problem on class distribution mismatch between labeled and unlabeled data in semi-supervised learning is important and realistic, (2) the application of doubly robust estimation is theoretically well-motivated, (3) the improvement on several benchmarks is modest but consistent. The main weaknesses are that (1) the method relies on strong assumptions and may violate in practice, indicating less empirical contributions, (2) the novelty of applying doubly robust retimator into SSL with a two-stage retraining framework is limited, (3) the improvement is modest but the computational cost is significant due to the second stage design, (4) the scale of benchmarks are not large enough.

The paper received four negative ratings and the authors did not provide a rebuttal. Considering the concerns that have not been addressed, the final rating is reject.

**Reviewer Concerns:**

The major concerns are on the strong assumptions and limited novelty. The authors did not provide a rebuttal to address the concerns.

**Reviewer Scores:**

As the authors did not provide a rebuttal, the scores would not increase.

---

### Decision · Program_Chairs · 2026-01-26

Reject